# Influence of Light Intensity and Photoperiod on the Photoautotrophic Growth and Lipid Content of the Microalgae *Verrucodesmus verrucosus* in a Photobioreactor

**Laura Vélez-Landa [1], Héctor Ricardo Hernández-De León [1], Yolanda Del Carmen Pérez-Luna [2], Sabino Velázquez-Trujillo [1], Joel Moreira-Acosta [3], Roberto Berrones-Hernández [2] and Yazmin Sánchez-Roque [2,\*]**

[1] Research Laboratory, Instituto Tecnológico de Tuxtla Gutiérrez, Carretera Panamericana KM. 1080, Tuxtla Gutiérrez C.P. 29050, Mexico; laura.velez@ittuxtlagutierrez.edu.mx (L.V.-L.); hhernandezd@ittg.edu.mx (H.R.H.-D.L.); sabinovelazquez1@gmail.com (S.V.-T.)

[2] Research Laboratory, Universidad Politécnica de Chiapas, Carretera Tuxtla-Villaflores KM. 1+500, Las Brisas, Suchiapa C.P. 29150, Mexico; yperez@upchiapas.edu.mx (Y.D.C.P.-L.); rberrones@upchiapas.edu.mx (R.B.-H.)

[3] Research Laboratory, Universidad del Valle de México, Tuxtla Gutiérrez, Chiapas , C.P. 29056, México; jmoreira23@yahoo.com.mx

\* Correspondence: ysanchez@ia.upchiapas.edu.mx; Tel.: +52-961-667-4365; Fax: +52-961-328-2021

**Abstract:** Microalgal biomass has the capacity to accumulate relatively large quantities of triacylglycerides (TAG) for the conversion of methyl esters of fatty acids (FAME) which has made microalgae a desirable alternative for the production of biofuels. In the present work *Verrucodesmus verrucosus* was evaluated under autotrophic growth conditions as a suitable source of oil for biodiesel production. For this purpose BG11 media were evaluated in three different light:dark photoperiods (L:D; 16:08; 12:12; 24:0) and light intensities (1000, 2000 and 3000 Lux) in a photobioreactor with a capacity of three liters; the evaluation of the microalgal biomass was carried out through the cell count with the use of the Neubauer chamber followed by the evaluation of the kinetic growth parameters. So, the lipid accumulation was determined through the lipid extraction with a Soxhlet system. Finally, the fatty acid profile of the total pooled lipids was determined using gas chromatography-mass spectroscopy (GC-MS). The results demonstrate that the best conditions are a photoperiod of 12 light hours and 12 dark hours with BG11 medium in a 3 L tubular photobioreactor with 0.3% $CO_2$, 25 °C and 2000 Lux, allowing a lipid accumulation of 50.42%. Palmitic acid is identified as the most abundant fatty acid at 44.90%.

**Keywords:** photobioreactor; light intensity; microalgal biomass; *Verrucodesmus verrucosus*

## 1. Introduction

Biofuel production and its study is an issue of global importance due to ever-increasing concerns over the depletion of fossil fuels and their devastating impact on the environment, so at present, more attention has been given to the production of biodiesel [1,2] which is a liquid biofuel, obtained from a transesterification process of an oil or fat with an alcohol [3]. The most common raw materials used for the production of biodiesel have been the basic vegetable oils, however, this presents multiple problems such as competition with the use of basic oils for nutritional purposes, the use of agricultural land, the use of pesticides and fertilizers necessary for crops, which in turn can leach out and contaminate the groundwater [2,4]. One potential alternative renewable raw material for the production of lipids to produce biodiesel are microalgae, which have the potential to overcome many limitations, they are capable of satisfying the world demand for transportation fuels [5] and they seem to be the only source able to completely displace diesel fossil [6]. Microalgae are photosynthetic microorganisms [7] with simple growth requirements that can produce around 80% lipid by dry mass weight, although the levels commonly obtained are 20–50% [8].

Unlike terrestrial crops, microalgae grow 10–50 times faster [5], which leads to higher biofuel volumes per hectare, for example the yield of microalgae oil (58,700 L ha$^{-1)}$ is about 31 times higher than that of jatropha (1892 L ha$^{-1}$) [2]. For the cultivation of microalgae, there are different systems [9], which use light (as an energy source), water and carbon dioxide (as a carbon source) to cultivate microorganisms, these in turn perform photosynthesis to generate biomass and oxygen [10]. Therefore, illumination factors, such as light intensity [11], photoperiod duration [12] and wavelength play an important role in the process of photosynthesis and affect photoautotrophic growth [13] and lipid content [14] of microalgae.

Specifically, photobioreactors currently use artificial light to characterize and optimize the living conditions of a microalgal morphotype. The most widely used are light emitting diodes (LEDs) that have become the currently preferred artificial light source in most phototrophic cultivations because of their durability (lifetime > 25–50,000 h) and lack of toxic elements such as mercury, unlike fluorescent lamps [15]. LEDs have also the advantage of being fast-responding diodes emitting nearly mono- or multichromatic light at desired wavelengths, thus being ideal for studies on the light requirements of microalgae for growth and light-dependent induction of specific target metabolites, without forgetting the importance of the times of exposure to light, called photoperiod [16]. Phototrophic cultivation benefits from the high power conversion efficiencies of LEDs, which can transform more than 40–50% of the electrical energy into light that can be effectively utilized by phototrophs for photosynthesis. However, it is important to consider the optimal specificity of each microalgae to obtain the metabolites of interest, which much be evaluated in each particular case [17].

Due to the aforementioned facts, microalgae have metabolic plasticity that allows them to adapt to different ecosystems and biotechnological processes that must be characterized. So far it is known that the main variables that determine the accumulation of lipids in the class Chlorophyceae are the light, aeration, pH, light type, photoperiod and temperature, with reported average values of 1500 Lux, 5% $CO_2$, 7.9, blue light at 475 nm, 16 L: 08 D and 22 °C, respectively. This class of microalgae has been reported to afford the highest levels of accumulated lipids [1,9,11,13,16,17].

For this reason, this work focuses on investigating the effect of light intensity and photoperiod on the growth of the microalga *Verrucodesmus verrucosus* using as a light source blue LED (465–470 nm), in a batch culture in a column photobioreactor design with a volume of 3 L.

## 2. Materials and Methods

### 2.1. Experimental

The experiment was set up at the Universidad Politécnica de Chiapas, located in the city of Suchiapa (Chiapas, Mexico). The geographic location is latitude 16°45′11″ north and longitude 93°06′56″, corresponding to the tropical region, with more than 1100 mm of annual rainfall. During the experimental period the temperature of the greenhouse was maintained at 28 °C. The air relative humidity was maintained at 60–65% with the objective of evaluating the efficiency of *Verrucodesmus verrucosus* (Roll) in synthetic medium BG11 for the production of lipids destined for the production of biodiesel.

For the implementation of this research study, nine treatments were established considering three light levels (1000, 2000 and 3000 Lux), three photoperiods (16:08, 12:12 and 24:00), and constant air injection with the presence of $CO_2$ and $O_2$ at concentrations of 0.03% and 7.5% respectively. The control corresponded to the treatment evaluated in flasks under natural light with 12:12 photoperiod, considering the same conditions of culture medium, $O_2$, and $CO_2$. The treatments were evaluated in triplicate.

### 2.2. Algal Sample

The sample of the algae *Verrucodesmus verrucosus* was provided by the Laboratory of Applied Science at Universidad Autónoma Metropolitana Iztapalapa (UAM). The species

was recently described by Hegewald et al. [18] and Refs. [19,20] and with GenBank number JQ240289.

### 2.3. Culture Medium

The culture BG11 medium with the composition given in Table 1 was used, the pH was adjusted to 7.5, It was brought to final volume of 1 L at 100 °C [21] (Table 1).

**Table 1.** Chemical composition of the BG11 culture medium used as a substrate for the microalgal biomass.

| BG-11 Medium for Blue Green Algae | | Trace Metal Mix A5 | |
|---|---|---|---|
| $NaNO_3$ | 1.5 g | $H_3BO_3$ | 2.86 g |
| $K_2HPO_4 \cdot 3H_2O$ | 4 g | $MnCl_2 \cdot 4H_2O$ | 1.81 g |
| $K_2HPO_4$ | 3.05 g | $MoO_4$ | 0.018 g |
| $MgSO_4 \cdot 7H_2O$ | 7.5 g | $ZnSO_4 \cdot 7H_2O$ | 0.222 g |
| $CaCl_2 \cdot 2H_2O$ | 3.6 g | $NaMoO_4 \cdot 2H_2O$ | 0.39 g |
| $HOC(COOH)(CH2COOH)_2 \cdot H_2O$ | 0.6 g | $CuSO_4 \cdot 5H_2O$ | 0.05 g |
| $(NH_4)_5[Fe(C_6H_4O_7)_2]$ | 0.6 g | $COCl_2 \cdot 6H_2O$ | 49.4 mg |
| $Na_2Mg$ EDTA | 0.1 g | Distilled water | 1.0 L |
| $Na_2CO_3$ | 0.02 g | | |
| Trace metal mix A5 | 1.0 mL | | |
| Distilled water | 1.0 L | | |

This preparation of the BG11 medium allows a total nitrogen concentration of 1 mg $L^{-1}$, it is important to mention that all the treatments started with the same nitrogen concentration.

### 2.4. Photobioreactor Design and Culture Conditions

*Verrucodesmus verrucosus* was grown under three light intensities (1000, 2000 and 3000 Lux) in three different photoperiods (16:08, 12:12 and 24:00) in a bubble column photobioreactor with a volume of 3 L at a temperature of 25 °C. The photobioreactor consisted of a cylindrical glass vessel 30 cm long and 12.5 cm in diameter with an expanded polyurethane cap with three holes for air inlet, sampling and oxygen outlet. The air (with 0.03% $CO_2$) was injected continuously into the culture from the bottom of the container through a plastic diffuser. Strips LEDs SMD5050 blue (465–470 nm) of 12 V and 3 A were used as the light source. The LED strips were placed horizontally (spirally) inside a cylindrical plastic container measuring 30 cm long and 23.8 cm in internal diameter, this container was placed on the photobioreactor and the LED strips were at a distance approximately 5 cm of the culture (Figure 1), also the light intensity was adjusted manually with a voltage regulator LM2596 with display with a power of 15 W, 2 A. The variation of the photoperiod was made using a 24 h analog timer. The light intensity and $CO_2$ were measured with a light and color sensor GDX-LC and sensor $CO_2$ inPro 5500i respectively, connected via USB to a laptop using the Graphical Analysis 4 software (Vernier Software and Technology, Beaverton, OR, USA). Five mL of culture was taken for sampling in a 24 h interval. Each experiment was carried out in triplicate to guarantee the reproducibility of the results (Figure 1).

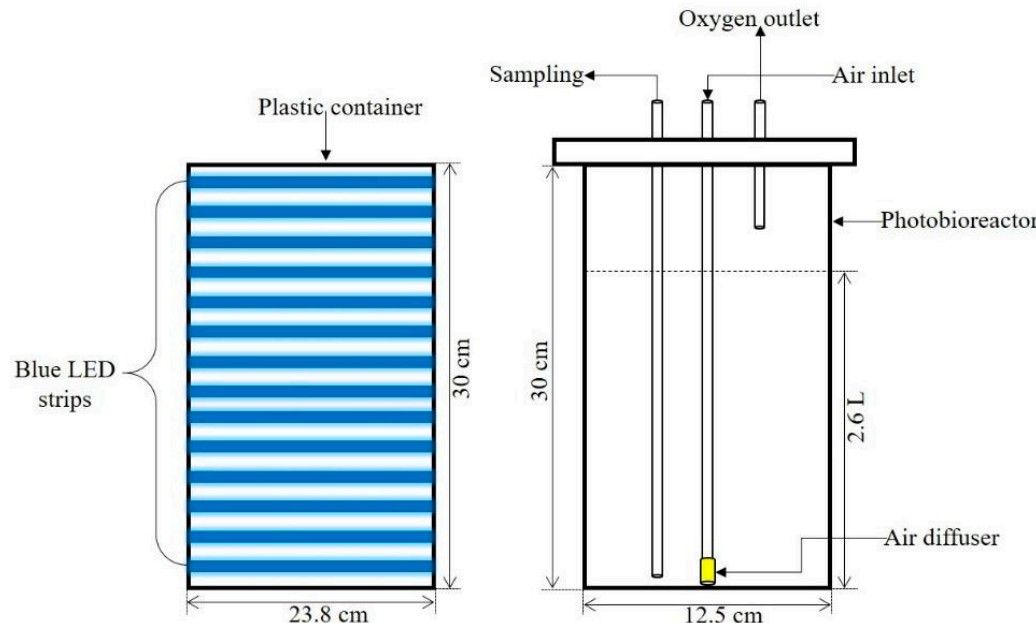

**Figure 1.** Design of a tubular photobioreactor with 3 L capacity for the production of microalgal biomass.

### 2.5. Growth Kinetics and Biomass Estimation

The microalgal species under study was characterized for growth kinetics and biomass production potential to identify those which grew fast and accumulate higher biomass. Cellular concentration was determined using a Neubauer hemacytometer (Hausser Scientific, Horsham, PA, USA) and a Primo Star microscope (Carl Zeiss, Oberkochen, Germany). The cells were harvested by centrifugation and dried at 70 °C. Growth rate parameters of each treatment were calculated on the basis of cell count during exponential phase of growth [22]. Following growth parameters were calculated from the observed data [23].

- Specific growth rate ($\mu$) was calculated from the equation: $\mu = \frac{\ln(N2-N1)}{t2-t1}$
- Where, $N2$ and $N1$ were the concentration of the number of cells at times $t2$ and $t1$ ($t2$ was considered the final time and $t1$ the beginning).
- Doubling time: ($Td$), the time required to double the number of cells was determined according to the equation: $Td = \frac{\ln 2}{\mu}$
- For the evaluation percentage of nitrogen consumption by microalgal biomass the total nitrogen (N) was analyzed by Kjeldahl method [23], considering the initial and final concentration.

### 2.6. Oil Extraction

The algae were oven dried at 70 °C, and after drying it was ground and sieved. Then 2 g of dried algae was placed in paper thimbles and that filter bag was loaded in the extraction chamber of a Soxhlet apparatus. The extraction chamber was kept over a boiling flask containing 50 mL of extraction solvent (hexane). The solvent was allowed to reflux at 60 °C for 45 cycles in 6 h, with a duration of 8 min per cycle.

The resulted lipids were removed from the oil by rotary evaporator to separate the lipids and n-hexane solvent for 3 h at 80 °C. After that, the lipid yield was weighed and calculated, for it the flask with the oil and without oil was weighed to determine the amount of oil obtained by weight difference [24].

### 2.7. Fatty Acid Profile of Total Lipids of Verrucodesmus verrucosus

Fatty acid profile analysis of species wise pooled total lipids isolated from *Verrucodesmus verrucosus* was carried out using gas chromatography-mass spectroscopy (GC-MS). Fatty acid methyl esters were prepared using the following procedure: 30 mg of total

lipid disolved in 1 mL of methanol was mixed with 1 mL of 12% solution of KOH prepared in methanol. To this solution equal volume of 5% HCl in methanol was added and heated at 75 °C for 15 min. This solution was allowed to cool and 1 mL of distilled water was added and shaken. Upper organic layer containing fatty acid methyl esters was carefully transferred to a new clean vial. GC-MS analysis of FAMEs was performed using diethylene glycol succinate capillary column (30 m × 0.25 × 0.25 μm). 100μL of methyl ester sample solution was injected for each analysis. Helium was used as a carrier gas. The injector temperature was 180 °C and detector temperature was 230 °C which was increased to 300 °C at a temperature gradient of 15 °C min$^{-1}$ [25].

### 2.8. Statistical Analysis

All experiments were performed in triplicate. Data are presented as mean ± SD. Statistical analysis was done using analysis of variance (ANOVA). *p*-values less than 0.05 were considered statistically significance.

### 3. Results

A tubular column photobioreactor with 3 L capacity was designed, using as a light source blue LED (465–470 nm) (Figure 1). During growth kinetics determination it was observed that the highest biomass production was reached at 2000 Lux in a 12:12 photoperiod with 6 × 106 cells (Figure 2). The highest specific growth rate was 1.2 ± 0.10 day$^{-1}$ and was obtained when the culture was exposed to a light intensity of 1000 Lux with a 16:08 h photoperiod, followed by the value 0.97 ± 0.00 day$^{-1}$ at 2000 Lux in a photoperiod of 24:00 h. In the three light intensities studied with a photoperiod of 12:12, the specific growth rate was lower compared to the other two photoperiod cycles (Table 2).

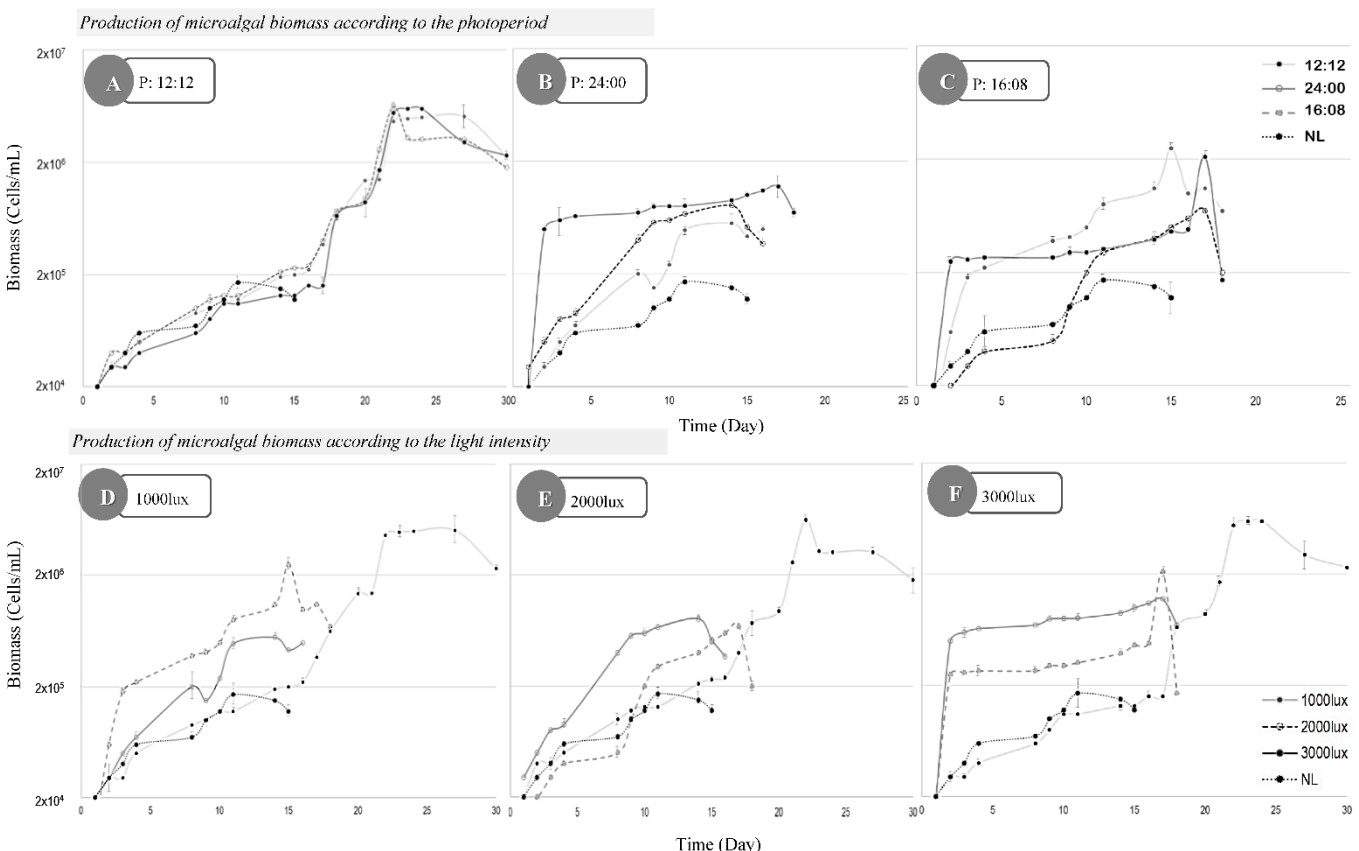

**Figure 2.** Growth kinetics of *Verrucodesmus verrucosus* in BG11 medium in three photoperiods [12:12; 24:00; 16:08; Natural light (NL)] for 30 days. Mean of three repetitions. The averages (±standard error) within each graphic.

**Table 2.** Kinetic growth parameters of *Verrucodesmus verrucosus* in BG11 medium in three photoperiods [12:12; 24:00; 16:08; Natural light (NL)] for 30 days.

| Light Intensity (Lux) | Photoperiod (Cycle L:D) | Initial Biomass (Cells · mL$^{-1}$) | Final Biomass (Cells · mL$^{-1}$) | μ (Day$^{-1}$) | Td (Day$^{-1}$) | Lipids (%) | Nitrogen Consumption (%) |
|---|---|---|---|---|---|---|---|
| | 16:08 | $2 \times 10^4 \pm 0.12$ [a] * | $2.50 \times 10^6 \pm 0.14$ [b] | $1.2 \pm 0.10$ [a] | $0.62 \pm 0.06$ [b] | $13.61 \pm 0.16$ [c] | $37 \pm 0.08$ [c] |
| 1000 | 12:12 | $2 \times 10^4 \pm 0.10$ [a] | $5.10 \times 10^6 \pm 0.06$ [a] | $0.50 \pm 0.09$ [b] | $1.39 \pm 0.20$ [a] | $42.41 \pm 0.11$ [a] | $55 \pm 0.11$ [b] |
| | 24:00 | $2 \times 10^4 \pm 0.21$ [a] | $5.60 \times 10^6 \pm 0.01$ [a] | $0.84 \pm 0.10$ [b] | $0.83 \pm 0.10$ [a] | $38.21 \pm 0.10$ [b] | $57 \pm 0.07$ [b] |
| | 16:08 | $2 \times 10^4 \pm 0.17$ [a] | $7.00 \times 10^5 \pm 0.02$ [b] | $0.70 \pm 0.00$ [a] | $0.99 \pm 0.00$ [b] | $37.35 \pm 0.09$ [c] | $19 \pm 0.02$ [d] |
| 2000 | 12:12 | $2 \times 10^4 \pm 0.14$ [a] | $6.30 \times 10^6 \pm 0.11$ [a] | $0.60 \pm 0.20$ [b] | $1.24 \pm 0.40$ [a] | $50.42 \pm 0.07$ [a] | $66 \pm 0.13$ [a] |
| | 24:00 | $2 \times 10^4 \pm 0.11$ [a] | $8.10 \times 10^5 \pm 0.13$ [b] | $0.97 \pm 0.00$ [a] | $0.72 \pm 0.00$ [c] | $43.09 \pm 0.6$ [b] | $21 \pm 0.21$ [d] |
| | 16:08 | $2 \times 10^4 \pm 0.07$ [a] | $4.80 \times 10^5 \pm 0.12$ [b] | $0.82 \pm 0.10$ [a] | $0.85 \pm 0.10$ [b] | $41.32 \pm 0.12$ [a] | $17 \pm 0.16$ [d] |
| 3000 | 12:12 | $2 \times 10^4 \pm 0.12$ [a] | $6.00 \times 10^6 \pm 0.09$ [a] | $0.58 \pm 0.09$ [b] | $1.21 \pm 0.20$ [a] | $29.29 \pm 0.06$ [c] | $62 \pm 0.12$ [a] |
| | 24:00 | $2 \times 10^4 \pm 0.18$ [a] | $1.20 \times 10^6 \pm 0.04$ [c] | $0.60 \pm 0.00$ [b] | $1.15 \pm 0.00$ [a] | $37.48 \pm 0.08$ [b] | $33 \pm 0.09$ [c] |
| Control | | $2 \times 10^4 \pm 0.21$ [a] | $1.70 \times 10^5 \pm 0.05$ [c] | $0.61 + 0.00$ [b] | $1.14 + 0.02$ [a] | $24.31 \pm 0.13$ [c] | $17.5 \pm 0.03$ [d] |

* Mean of three repetitions and the averages (±standard error) within each column. Different letters show statistically significant differences at $p < 0.05$. Td: Doubling time; μ: Growth rate; L:D: Light:Darkness.

On the other hand, it is important to mention that during the growth kinetics study a greater adaptation by the microalgae was observed in the 12:12 photoperiod, followed by the 24:00 and 16:08 photoperiod, so it was also possible to show that during the photoperiod 12:12 the microalgae biomass maintained an exponential phase of up to 23 days, unlike the 24:00 and 16:08 photoperiod, which maintained an exponential phase of 14 and 17 days respectively.

However, it is relevant to mention that the production of microalgal biomass in the photobioreactor of 3 L exceeds two logarithmic cycles compared to the flask production. At the beginning of the growth kinetics, it was observed that the microalgal biomass in the 12:12 photoperiod presented greater metabolic adaptation since the exponential phase began on the second day of the inoculation process unlike the other photoperiods that demonstrated a prolonged latency stage of 8 days for the start of the exponential stage (Figure 2).

With regards to lipid production, it was observed that during the flask evaluation, the maximum production was 24.31%, however a significant difference was observed during the evaluation in a photobioreactor at different light intensities and photoperiods, observing the best results at 2000 Lux and photoperiod 12:12 where the lipid concentration doubled (50.42%). This high concentration of lipids is related to the high consumption of nitrogen (66%) by the microalgal biomass (Table 2), On the other hand, the lowest production of lipid (13. 61%) was identified at 1000 Lux with photoperiod 16:08.

Regarding the chromatographic analysis of the profile of fatty acids present in the lipid samples obtained from the better yields at different light intensities, palmitic acid was observed as the most abundant of 42.03 to 55.21% (Figure 3). Of the identified fatty acids, 64.29% correspond to saturated fatty acids and 35.71% correspond to monounsaturated fatty acids (Table 3).

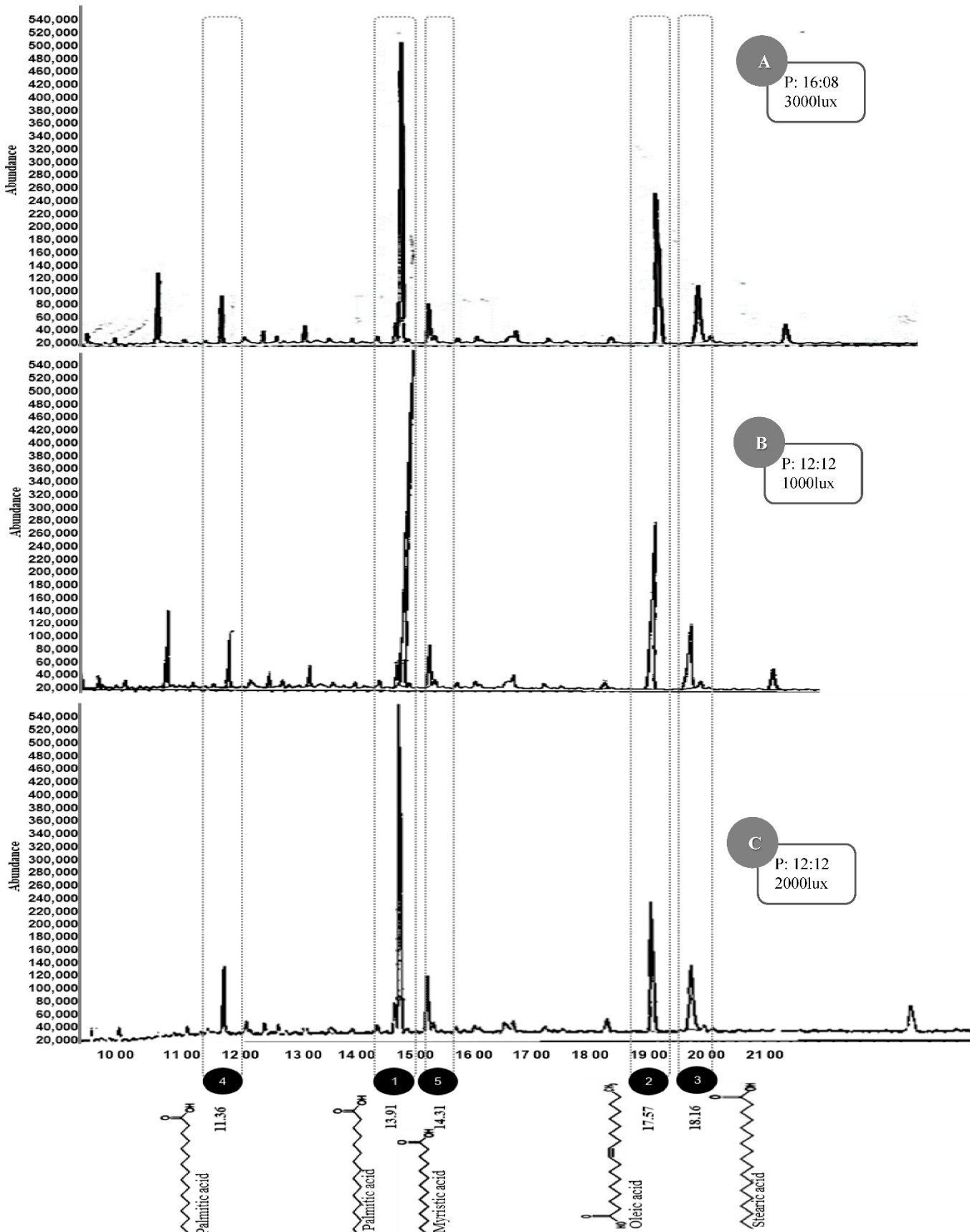

**Figure 3.** Chromatogram of fatty acid profile analysis from *Verrucodesmus verrucosus* in medium BG11 at 1000, 2000 and 3000 Lux, using gas chromatography mass spectroscopy (GC-MS).

**Table 3.** Identification of the compounds present in the chromatogram of fatty acid profile analysis from *Verrucodesmus verrucosus* in medium BG11 at 1000, 2000 and 3000 Lux, using gas chromatography mass spectroscopy (GC-MS).

| Compounds | Chemical Formula | Parent Ion (*m/z*) [2] | RT [1] (min) | Abundance Percentage (%) | | | Fatty Acid |
|---|---|---|---|---|---|---|---|
| | | | | *p* = 12:12 2000 Lux | *p* = 12:12 1000 Lux | *p* = 16:08 3000 Lux | |
| Palmitic acid | $C_{16}H_{32}O_2$ | 256.43 | 13.91 | 44.90 ± 0.05 [b] | 55.21 ± 0.12 [a] | 42.03 ± 0.01 [b] | Saturated |
| Oleic acid | $C_{18}H_{34}O_2$ | 282.47 | 17.57 | 25.81 ± 0.11 [b] | 30.01 ± 0.10 [b] | 28.05 ± 0.05 [b] | Monounsaturated |
| Stearic acid | $C_{18}H_{36}O_2$ | 284.48 | 18.16 | 9.81 ± 0.15 [a] | 9.55 ± 0.08 [a] | 9.76 ± 0.19 [a] | Saturated |
| Palmitoleic acid | $C_{16}H_{30}O_2$ | 254.41 | 11.36 | 7.87 ± 0.22 [a] | 7.00 ± 0.12 [a] | 5.50 ± 0.21 [a] | Monounsaturated |
| Myristic acid | $C_{14}H_{28}O_2$ | 228.36 | 14.31 | 5.93 ± 0.07 [a] | 4.95 ± 0.02 [a] | 4.35 ± 017 [a] | Saturated |

[1] RT: retention time, [2] Parent ion (*m/z*): molecular ions of the standard compounds (mass to charge ratio). Different letters show statistically significant differences at *p* < 0.05.

## 4. Discussion

According to the results obtained the photoperiod determines the lipid accumulation and biomass production. Conversions of phosphoenolpyruvate to pyruvate and 3-phosphoglycerate to glycerol 3-phosphate are the candidate rate-limiting steps responsible for delayed lipid accumulation [26]. The accumulation of substrates including ribulose 5-phosphate could be explained by the close relationship of increased biomass yield with enhanced $CO_2$ fixation, related to the dynamic metabolic profiling of lipid/carbohydrate anabolism (including production of 3-phosphoglycerate, fructose 6-phosphate, glucose 6-phosphate, phosphoenolpyruvate and acetyl-CoA) from $CO_2$ [27].

On the other hand, the influence of light/dark cycling on lipid accumulation and biomass production has been characterized in some microalgal species. As demonstrated in this research paper, where the best photoperiod for obtaining lipids was 12:12 (Table 1), these results are similar to those obtained in other research papers where they evaluated *Chlorella vulgaris,* and higher biomass production and lipid content were achieved under 12 h: 12 h light:dark cycles relative to a 24 h: 0 h continuous illumination condition [28]. In *Nannochloropsis sp.,* lipid content and growth rate were highest under an 18 h:6 h light/dark cycle compared with 24 h:0 h and 12 h:12 h cycles [12,22]. In *Nannochloropsis gaditana*, biomass concentration was highest under a 12 h:12 h light/dark cycle, while lipid content was highest under a 16 h:8 h cycle, relative to 24 h:0 h and 8 h:16 h cycles [9]. In *Tribonema minus*, however, biomass production and lipid content were higher under 24 h:0 h continuous illumination compared with that under a more natural 12 h:12 h cycle [29].

Also during the growth kinetics study it was observed that the highest biomass production was reached at 2000 Lux in a 12:12 photoperiod with 6 × 10⁶ cells and lipid production of 50.42% (Table 2, and Figure 2), on the other hand, the influence of light/dark cycling on lipid accumulation and biomass production has been characterized in some microalgal species, as was demonstrated in this research paper, where the best photoperiod was 12:12 for the lipids obtention with 1000 and 2000 Lux, not so for 3000 Lux wich does not differ statistically from the control (Table 2). At a light intensity of 3000 Lux the highest percentage of lipids corresponded to the 16:08 photoperiod, so increasing the light to improve biomass is not an option. Such an effect was previously demonstrated by Fathurrahman et al. [28] who mentioned that microalgae have an efficient conversion threshold of available light for biomass, this may be due to the fact that at higher light

intensity some species of microalgae show evidence of photodamage [23,29], so that the light intensity can become so high that it generates saturation or photoinhibition, causing microalgal growth inhibition [30]. However, it is important to mention that each microalgal morphotype has its own metabolic plasticity, so it is important to evaluate them and optimize variables such as light and photoperiod individually to know the metabolic efficiency in terms of lipid production, in this sense, the results indicate that the kinetic model can realistically reflect the light energy utilization efficiency of microalgae as well as their intrinsic growth kinetic characteristics [13,31]. The aforementioned results were detected because the microalgae naturally are exposed to changing light conditions, these light capture fuels all photosynthetically-driven microalgae processes, therefore, depending on the intensity of the light and the exposure time, the microalgae can undergo photoacclimation, reversible photodamage to photosystem II (PSII), and more severe photodamage to photosystem I (PSI) [32].

Regarding the accumulation of lipids Keil et al. [33] and Mutaf et al. [34] mentioned that green algae can accumulate lipids containing 12.05 to 71.33% of fatty acids. The high concentration of lipids in green microalgae was related to the physicochemical composition of the substrate, so in this research work the BG11 medium was used according to Sánchez et al. [35]; this culture medium contains a concentration of nitrogen of less than 2 mg/L, a concentration equivalent to that used by Xin et al. [36] who mentioned that under conditions of nitrogen ($2.5$ mg $L^{-1}$) or phosphorus ($0.1$ mg $L^{-1}$) limitation *Scenedesmus* sp. could accumulate up to 53% lipids [37–41].

In the present research work it was observed that 64.29% of the fatty acids composition present corresponded to saturated fatty acids (Table 2), as demonstrated by Tan et al. [42] who observed that the quality of biodiesel improved to greater saturation of fatty acids (typically in C16:0 and C18:0). The composition of saturated fatty acids in high concentration in green microalgae such as *V. verrucosus* is due to the low presence of the enzyme diacylglycerol acyltransferase responsible for the biosynthesis of neutral lipids [33].

Other researchers who have evaluated various microalgae corresponding to the Chlorophyceae class, specifically *Scenedesmus* sp. which is part of the Scenedesmaceae family like *V. verrucosus*, have reported that this morphotype accumulated 35–50% of lipids. On the other hand, lipid content was higher than *Ettlia sp, Chlorella vulgaris, Neochloris oleabundans, Nannochloropsis* sp., *N. salina, N. oceanica, N. oculata* and *Scenedesmus dimorphus*. Palmitic acid was identified as the most abundant fatty acid at 40%. This result indicated that it can be a promising feedstock for biodiesel production, because the most common components of biodiesel are palmitic acid, palmitoleic acid and oleic acid [35,43–45].

## 5. Conclusions

In the present research work, the metabolic plasticity of microalgae is corroborated, demonstrating that the conditions to which they are exposed determine the accumulation of lipids, and confirming the potential of green microalgae corresponding to the Chlorophyceae class as the group that has reported greater concentration of lipids. In this sense the results showed that the *Verrucodesmus verrucosus* microalgae has better growth and adaptation in a 3 L tubular photoreactor under controlled conditions at a light intensity of 2000 Lux and a 12 L: 12 D photoperiod, with a total lipid accumulation of 50.42% with fatty acids such as palmitic acid, oleic acid, stearic acid, palmitoleic acid and myristic ac-id in concentrations of 44.90, 25.81, 9.81, 7.87 and 5.93%, respectively.

However, it is important to mention that each microalga is specific, so it requires a previous evaluation that allows the optimization of the conditions to potentiate the production of lipids that can be used for conversion to biodiesel. This specificity turns out to be very promising for future research with morphotypes that, in addition to achieving high yields, it can be produced at low cost.

**Author Contributions:** Conceptualization, L.V.-L. and Y.S.-R.; methodology, H.R.H.-D.L. and R.B.-H.; software, L.V.-L.; validation, Y.D.C.P.-L.; formal analysis, S.V.-T. and J.M.-A.; investigation, and writing original draft preparation, L.V.-L. and Y.S.-R.; writing—review and editing. All authors have read and agreed to the published version of the manuscript.

**Funding:** This research was supported by Instituto Tecnológico de Tuxtla Gutiérrez and Universidad Politécnica de Chiapas.

**Institutional Review Board Statement:** Not applicable.

**Informed Consent Statement:** Not applicable.

**Data Availability Statement:** Please contact corresponding author. The data used in this paper are the result of objective experimental work under strict scientific evaluation.

**Acknowledgments:** The autors gratefully acknowledge to the UPCH and ITTG for financial support.

**Conflicts of Interest:** The authors declare no conflict of interest.

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
