# Peer review of "Influence of Light Intensity and Photoperiod on the Photoautotrophic Growth and Lipid Content of the Microalgae Verrucodesmus verrucosus in a Photobioreactor"

_sustainability, doi:10.3390/su13126606_

Round 1
Reviewer 1 Report
The introduction is very generic. No background information is provided on the algae species of the current study, on previous relevant studies etc. Moreover, the scope of the study and the novelty are not provided at al. Please expand the section based on the above comments.
Lines 59-60: I cannit see the relevance of this argument with the content of the previous and later comments.
Line 79: What do you mean by” Immediately, it was brought to laboratory “. Immediately after what?
Line 79 and elsewhere I the text: Please use “species” instead of “specie”. In both the original Latin and in English “species” is the spelling of both the singular and plural forms. Specie is a technical term referring to the physical form of money, particularly coins.
Lines 79-84: Morphological similarity cannot be a guarantee for the taxonomic identification. I visited the sites you refer, too and the methodology you followed is not apparent. The identification could only be considered precise if made by molecular tools. If this is not feasible, it would be preferable if you provide evidence on the methodology you followed for the morphological identification of the alga incorporating it in the results section and refer to the alga as newly isolated strain bearing the characteristics of Verrucodesmus verrucosus. I would also recommend not to use the alga name in the title of the study.
Line 98: Why did you use air and not CO2? What was the air supply rate in the medium? Was the
Line 119: Was drying of the cell performed in order to estimate biomass concentration in terms of g/L? Please explain.
Line 130: What is meant by ‘filter bag”. Do you refer to the paper thimbles?
Line 134: What was the retention time and how many cycles did you perform? In case you used a standard protocol please provide the exact reference. In case you used a modified protocol or a new protocol that you developed, please describe it in detail.
Line 139: Please rephrase, the meaning is not clear
Figure 2: the starting point for the y axis of fig.2a is 2.104, whereas in figs2b and c it is 0. Why is that? Please use the same scale in the y axis of all three graphs for comparison reasons. It would also be useful for comparison reasons to provide a graph with the variation of light intensity changes using NL for each set of experiments? Are such data available?
Lines 157-174 and figure 2: What was the nitrogen source consumption rate and trend in each case? Did you quantify that? I guess that CO2 sparging was the same in all reactors and as such the availability of carbon was the same. As such the limiting substrate would be the NO3. It would also be useful to estimate the total nitrogen consumption in each case and based on that the biomass yield YX/S
Line 170: you refer to the growth rates in flask experiments, but no mention is made on those experiments in the materials and section method. What were the conditions of those experiments (medium, Xo, CO2 rate, illuminance, volumes etc) and the results? I am afraid that there is no point in making comparisons if the data are not provided.
Line 179: Please see the previous comment.
Lines 182-188 and table 1: Please provide the SE for the estimated lipid percentage values.
Figure 3: the figure can be omitted since the exact same information is provided in table 1
Lines 185-188: It would be interesting to compare the changes of the distribution of fatty acid during the different handling.
Lines 223-226: Is that the case for the specific alga , too?
Table 2: the table can be omitted since the same information is provided in figure 4
Lines 234-237: Intracellular accumulation of carbon sources in the form of lipids, polysaccharides and polyesters is common for many species under nutrient limiting concentration or environmental stress. There are microorganisms though that perform accumulation during growth, what is the case of V. verrucosus? Please refer to previous studies of the same alga. Moreover it is important to quantify nitrogen in your medium at different stages of the microbial growth and also estimate lipid content at the same time so as to evaluate the effect of nitrogen efficiency on the accumulation.
Lines 231-237: The information provided here are very generic. It would be useful to expand the discussion with regard to previous studies of V. verrucosus.
Author Response
Tuxtla Gutierrez Chiapas, Mexico, 25-04-2021
Dear Prof. Dr. Marc A. Rosen
Editor-in-Chief
Sustainability
We would like to submit the revised manuscript entitled ‘Influence of Light Intensity and
Photoperiod on the Photoautotrophic Growth and Lipid Content of the Microalgae
Verrucodesmus verrucosus in a Photobioreactor’ by Laura Velez et al. for publication in Sustainability. We corrected the manuscript and included all suggestions and comments made by the editor in the revised manuscript. Please find below the comments by the reviewers in italic, our answers in normal font and the changes to the text in red color.
We hope this makes the revised manuscript acceptable to be published in Sustainability.
Hoping to hear from you soon
Sincerely yours,
Dra. Yazmin Sánchez Roque
On behalf of the co-authors
Universidad Politécnica de Chiapas
e-mail: ysanchez@ia.upchiapas.edu.mx
_____________________________________
Manuscript 1192551 “Influence of Light Intensity and Photoperiod on the Photoautotrophic Growth and Lipid Content of the Microalgae Verrucodesmus verrucosus in a Photobioreactor”; Sustainability
Reviewer 1
- The introduction is very generic. No background information is provided on the algae species of the current study, on previous relevant studies etc. Moreover, the scope of the study and the novelty are not provided at al. Please expand the section based on the above comments.
We appreciate the editor's comments, so, we strengthened the introduction with the following wording.
Line 62-72 “Microalgae have metabolic plasticity that allows them to adapt to different ecosys-tems and biotechnological processes, generating biomass that can be used in the produc-tion of food, concentrates, bioactive compounds, in bioremediation and biofuels. Since light and temperatur are the main determinants of biomass productivity of crops in open-air ponds operated under well-mixed basic nutrients (N, P, CO2), it is necessary to know the specific growt rate of the strain selected as a function of these two variables [7,8].
In other studies, Verrucodesmus verrucosus allowed the highest accumulation of lipids with 24.3% in a period of 60 days and a 12:12 photoperiod at the laboratory level, however it allows the accumulation of 50% of lipids in a period of 15 days and 12:12 photoperiod at 2000 lux in a tubular photobioreactor, said accumulation is achieved with an adequate nitrogen concentration of 0.23% [24].
For this reason, in this work focuses on investigating the effect of light intensity and photoperiod on the growth of the microalga Verrucodesmus verrucosus using as a light source blue LED (465-470 nm), in a batch culture in a column photobioreactor design with volume of 3 L.”.
- Lines 59-60: I cannit see the relevance of this argument with the content of the previous and later comments.
We agree with the reviewer, we remove this wording and we generate an adjustment in the wording:
Line 56. “which use light (as an energy source), water and carbon dioxide (as a carbon source) to cultivate microorganisms”.
- Line 79: What do you mean by” Immediately, it was brought to laboratory “. Immediately after what?
We appreciate the reviewer's observation, we improve the wording.
- Line 79 and elsewhere I the text: Please use “species” instead of “specie”. In both the original Latin and in English “species” is the spelling of both the singular and plural forms. Specie is a technical term referring to the physical form of money, particularly coins.
We agree with your observation, we review the document and correct the term.
- Lines 79-84: Morphological similarity cannot be a guarantee for the taxonomic identification. I visited the sites you refer, too and the methodology you followed is not apparent. The identification could only be considered precise if made by molecular tools. If this is not feasible, it would be preferable if you provide evidence on the methodology you followed for the morphological identification of the alga incorporating it in the results section and refer to the alga as newly isolated strain bearing the characteristics of Verrucodesmus verrucosus. I would also recommend not to use the alga name in the title of the study.
We appreciate the editor's comments, so, we changed the wording and specified the origin of the identification of the strain used:
Line 97-101 “The algae sample Verrucodesmus verrucosus was provided by the laboratory of applied science in Universidad Autónoma Metropolitana Iztapalapa (UAM). The species was re-cently described by Hegewald et al. (2013); López-Mendoza et al. (2015) and with Gen-Bank number JQ240289”.
- Line 98: Why did you use air and not CO2? What was the air supply rate in the medium? Was the
We appreciate the editor's comments, however, the reason air was included instead of pure CO2 is due to the following:
The CO2 is supplies the culture mixed with air, due to, in addition to serving as the main source for microalgal nutrition, it represents the control method and maintenace of optimal pH levels for crops, likewise, the use of air reduces the costs of producing microalgal biomass in a photobioreactor for the accumulation of lipids.
- Line 119: Was drying of the cell performed in order to estimate biomass concentration in terms of g/L? Please explain.
We appreciate your observation, however in the methodology (line 138-144) it is indicated that the growth kinetics of the microalgal biomass was carried out with the neubauer chamber, so this result is reflected in figure 2 in units of cells / mL and the drying process was used for the extraction of lipids in the soxtlet equipment, this result allowed the calculation of yields
- Line 130: What is meant by ‘filter bag”. Do you refer to the paper thimbles?
We agree with the reviewer, it effectively refers to the paper thimbles, in this sense the word was changed in the text (line 130).
156 “paper thimbles”
- Line 134: What was the retention time and how many cycles did you perform? In case you used a standard protocol please provide the exact reference. In case you used a modified protocol or a new protocol that you developed, please describe it in detail.
We agree with the reviewer, therefore we restructure the lipid extraction methodology:
Line 155-163. “The algae was oven dried at 70 ºC, after drying it was ground and sieved. Then 2 g of dried algae was tied in paper thimbles and that filter bag was loaded in extraction cham-ber of Soxhlet apparatus. The extraction chamber was kept over a boiling flask containing 50 mL of extraction solvent. Hexane was used as solvent. The solvent was allowed to re-flux at 60 ºC for 45 cycles in 6h, with a duration of 8min per cycle.
The resulted lipids were removed from the oil by rotary evaporator to separate the li-pids and n-hexane solvent for 3 hours at 80°C. After that, the lipid yield was weighed and calculated, for it the flask with the oil and without oil was weighed to determine the amount of oil obtained by weight difference [29, 30, 31]”.
- Line 139: Please rephrase, the meaning is not clear
We agree with the reviewer, therefore we restructure the methodology as mentioned previously in number 9.
- Figure 2: the starting point for the y axis of fig.2a is 2.104, whereas in figs2b and c it is 0. Why is that? Please use the same scale in the y axis of all three graphs for comparison reasons. It would also be useful for comparison reasons to provide a graph with the variation of light intensity changes using NL for each set of experiments? Are such data available?
We appreciate the reviewer's observations, we totally agree, therefore the redesign of figure 2 was carried out considering the analysis of the axes and including the suggested graphs depending on the light conditions.
- Lines 157-174 and figure 2: What was the nitrogen source consumption rate and trend in each case? Did you quantify that? I guess that CO2 sparging was the same in all reactors and as such the availability of carbon was the same. As such the limiting substrate would be the NO3. It would also be useful to estimate the total nitrogen consumption in each case and based on that the biomass yield YX/S
We appreciate the editor's comments, so, we included in the methodology (line 115) the concentration of CO2 that was injected into the photobioreactor continuously, as well as the technique used for the evaluation of total nitrogen (line 146-148), at the beginning and end of the growth kinetics, considering that all treatments start with the same nitrogen concentration provided by the BG11 medium (line 107-109).
Finally, in response to his comment, we include the percentage of nitrogen consumption by the microalgal biomass.
Line 120. “. The air (with 0.03% CO2) was injected continuously into the culture from the bottom of the container through a plastic diffuser”.
Line 151-153. “For the evaluation percentage of nitrogen consumption by microalgal biomass the total ni-trogen (N) was analyzed by Kjeldahl method [24], considering the initial and final concentration”
Line 112-114. “This preparation of the BG11 medium allows a total nitrogen concentration of 1 mg L-1, it is important to mention that all the treatments started with the same nitrogen concen-tration”.
- Line 170: you refer to the growth rates in flask experiments, but no mention is made on those experiments in the materials and section method. What were the conditions of those experiments (medium, Xo, CO2 rate, illuminance, volumes etc) and the results? I am afraid that there is no point in making comparisons if the data are not provided.
We agree with the reviewer, therefore, we have improved the wording and have included the information requested in the methodology (Line 86-92).
Line 86-92. “For the implementation of this research work, nine treatments were established con-sidering three light levels (1000, 2000 and 3000lux), three photoperiods (16:08, 12:12 and 24:00), constant air injection with the presence of CO2 and O2 at concentrations of 0.03% and 7.5% respectively, finally to point out that the control corresponds to the treatment evaluated in flasks under natural light with 12:12 photoperiod, considering the same con-ditions of culture medium, O2, and CO2. The treatments were evaluated in triplicate”.
- Line 179: Please see the previous comment.
We agree with the reviewer, in this sense, with the previous intervention we consider the wording understandable.
- Lines 182-188 and table 1: Please provide the SE for the estimated lipid percentage values.
We agree with the reviewer, therefore, we have included the standard deviation in the values referring to the percentage of lipids.
- Figure 3: the figure can be omitted since the exact same information is provided in table 1
We agree with the reviewer, therefore, we have eliminated figure 3.
- Lines 185-188: It would be interesting to compare the changes of the distribution of fatty acid during the different handling.
We value the reviewer's observations and in response to their suggestions, we include the chromatograms corresponding to the 3 treatments with the highest concentrations of lipids under the three light intensities, considering the central objective of this research work (Figure 3).
- Lines 223-226: Is that the case for the specific alga, too?
We appreciate the reviewer's observation, however, this information relates to the results presented in Table 2 and the references used, since they mention this behavior in microalgae corresponding to the same class (chlorophyceas).
- Lines 234-237: Intracellular accumulation of carbon sources in the form of lipids, polysaccharides and polyesters is common for many species under nutrient limiting concentration or environmental stress. There are microorganisms though that perform accumulation during growth, what is the case of V. verrucosus? Please refer to previous studies of the same alga. Moreover it is important to quantify nitrogen in your medium at different stages of the microbial growth and also estimate lipid content at the same time so as to evaluate the effect of nitrogen efficiency on the accumulation.
We appreciate the reviewer's observations, however V. verrucosus is a recent species, so it has shown a lot of evidence in bioremediation studies but little research for this morphotype in the area of bioenergies, so we consider it pertinent to continue future evaluations with this microalgae. So far, the most important findings are as follows:
In a investigation work Verrucodesmus verrucosus was evaluated in mixed and autotrophic growth conditions as a suitable source of oil for biodiesel production. So to, the nitrogen-lipid ratio was determined and lipid extraction. The results demonstrate that the maximum cell density was reached in the Guillard medium with 6.0 x 104 cells / mL. On the other hand, the results in the production of lipids of the microalgal biomass showed that the BG11 medium is the best, since it produces up to 24.3% of lipids, finally, the Fatty acid profile shows that the highest concentrations of fatty acids in V. verrucosus were palmitic, oleic, stearic, with a concentration of 34.9, 22.8 and 9.3% respectively, indicating the suitability of this species for biodiesel production.
- Lines 231-237: The information provided here are very generic. It would be useful to expand the discussion with regard to previous studies of V. verrucosus.
We appreciate the reviewer's observations, however V. verrucosus is a recent species in the area of bioenergies, so we consider it pertinent to continue with future evaluations with this microalgae. So far, the most important findings are those mentioned above by (Sánchez-Roque 2020).

Reviewer 2 Report
The article and its content is interesting and important in the context of searching for raw materials and processes for obtaining fuels and generating energy that are environmentally friendly. This topic (cultivation of microalgae) is now widely undertaken by researchers around the world. The obtained and presented data are interesting, but the current way of presenting them requires rewording to make it more precise, clear and approcheable to the reader . In general, the discussion of the results in this article should be more extensive in terms of all parameters assessed by the authors, and discussed in more detail in the context of the literature data on this subject. This also applies to the conclusions made by the authors, which are very perfunctory. Therefore I suggest major revision. More detailed comments are included below:
Subsection 2.3. - for clarity of text, the composition of culture medium should be tabulated.
Interpretation of the results in Figure 2 presents problems for the reader. In the 16:08 and 24:00 photoperiods it is more readable, while at 12:12 at first glance, the amount of biomass produced (cell/mL) is quite similar, regardless of the light intensity - of course, this is due to the logarithmization of the Y axis which makes it difficult to read a specific value. Therefore it is proposed that all data presented in Figure 2 also be included in the form of a table (for example in the appendix). If possible, it would be more advantageous to present the data in mg / L and without the logarithmic Y axis and with plotted error bars. In the table 1, the authors included only the initial and final amount of biomass (and in fact the maximum obtained). These are insufficient data for a proper evaluation of the results by the reader.
Some errors are also in table 1: it should be Cell instead of Cél; for doubling time is an error in the unit; Table 1 also shows the lipid content, which is important for the production of biodiesel - but there is no information about the standard deviation.
In subsection 2.8 the authors reported about the performed analysis of variances ANOVA , but they have not presented their results anywhere. There are only some indexes in Table 1, which say whether the average data are significantly different - the results of the analysis should be attached, for example, in the appendix.
In line 222 the authors suggest that an increase in light intensity causes a decrease in Lipid content. This conclusion is debatable (which does not mean that it is not true) but according to the data presented by the authors in Table 1 it is true only for the 12:12 photoperiod - significant decrease was noted for 3000 Lux - the differences between 1000 and 2000 are insignificant (according to the indices described by the values, no results of the variance analysis) - but the authors do not mention it. This requires a broader discussion. This also applies to the other assessed parameters, the changes and trends of which, along with the change in the intensity of light and photoperiods, have been discussed little in depth. With regard to the presented data, it is suggested to combine the results and discussion sections into one section.
Moreover, there are editorial errors: In line 158 and 219 it should be 6 x 106 cells/ml instead of 6X106 cells; in various places in the text, the unit of light intensity is lux, lx and Lux; ml and mL. The text should be thoroughly revised and errors of this type should be corrected
Also, Figure 3 is not very clear - I suggest modifying it and adding standard error bars. The description is also about averages (± standard error) - no such data in the graph. In the graph, the colors represent the photoperiods, and the legend on the right adds no additional information.
Author Response
Tuxtla Gutierrez Chiapas, Mexico, 25-04-2021
Dear Prof. Dr. Marc A. Rosen
Editor-in-Chief
Sustainability
We would like to submit the revised manuscript entitled ‘Influence of Light Intensity and
Photoperiod on the Photoautotrophic Growth and Lipid Content of the Microalgae
Verrucodesmus verrucosus in a Photobioreactor’ by Laura Velez et al. for publication in Sustainability. We corrected the manuscript and included all suggestions and comments made by the editor in the revised manuscript. Please find below the comments by the reviewers in italic, our answers in normal font and the changes to the text in red color.
We hope this makes the revised manuscript acceptable to be published in Sustainability.
Hoping to hear from you soon
Sincerely yours,
Dra. Yazmin Sánchez Roque
On behalf of the co-authors
Universidad Politécnica de Chiapas
e-mail: ysanchez@ia.upchiapas.edu.mx
______________________________________
Reviewer 2
- Subsection 2.3. - for clarity of text, the composition of culture medium should be tabulated.
We agree with the reviewer, therefore we have represented the chemical composition of the BG11 medium in a table
- Interpretation of the results in Figure 2 presents problems for the reader. In the 16:08 and 24:00 photoperiods it is more readable, while at 12:12 at first glance, the amount of biomass produced (cell/mL) is quite similar, regardless of the light intensity - of course, this is due to the logarithmization of the Y axis which makes it difficult to read a specific value. Therefore it is proposed that all data presented in Figure 2 also be included in the form of a table (for example in the appendix). If possible, it would be more advantageous to present the data in mg / L and without the logarithmic Y axis and with plotted error bars. In the table 1, the authors included only the initial and final amount of biomass (and in fact the maximum obtained). These are insufficient data for a proper evaluation of the results by the reader.
We agree with the reviewer, therefore figure 2 has been restructured in order to facilitate the interpretation of the results.
- Some errors are also in table 2: it should be Cell instead of Cél; for doubling time is an error in the unit; Table 2 also shows the lipid content, which is important for the production of biodiesel - but there is no information about the standard deviation.
We agree with the reviewer, therefore, the correction of the translation error (Table 2) was addressed and the standard deviation of the results of the total lipid content per treatment was included (Table 2).
- In subsection 2.8 the authors reported about the performed analysis of variances ANOVA, but they have not presented their results anywhere. There are only some indexes in Table 1, which say whether the average data are significantly different - the results of the analysis should be attached, for example, in the appendix.
We appreciate the reviewer's observation; however, the handling of letters has been included in the different results of the variables evaluated in this research work, for which it refers to the development of a statistical analysis with the central objective of identifying significant differences.
- In line 222 the authors suggest that an increase in light intensity causes a decrease in Lipid content. This conclusion is debatable (which does not mean that it is not true) but according to the data presented by the authors in Table 1 it is true only for the 12:12 photoperiod - significant decrease was noted for 3000 Lux - the differences between 1000 and 2000 are insignificant (according to the indices described by the values, no results of the variance analysis) - but the authors do not mention it. This requires a broader discussion. This also applies to the other assessed parameters, the changes and trends of which, along with the change in the intensity of light and photoperiods, have been discussed little in depth. With regard to the presented data, it is suggested to combine the results and discussion sections into one section.
We agree with the reviewer, therefore, we have restructured the wording and have implemented other elements to the discussion, with the aim of facilitating the analysis by the reader.
The change is observed of the line 301 to 320: So to during growth kinetics it is observed that the highest biomass production was reached at 2000 Lux in a 12:12 photoperiod with 6X106 cells and lipid production of 50.42% (Table 1, and Figure 3), on the other hand, the influence of light/dark cycling on li-pid accumulation and biomass production has been characterized in some microalgal species, as demonstrated in this research paper, where the best photoperiod was 12:12 for the lipids obtention with 1000 and 2000 lux, not so for 3000 lux wich does not differ statistically from the control (Table 1). At a light intensity of 3000 lux the highest per-centage of lipids corresponded to the 16:08 photoperiod. So, Increasing the light to im-prove biomass is not an option, such an effect was demonstrated by Fathurrahman et al. [37] when mentioning that microalga have an efficient conversion threshold of available light for biomass, this may be due to the fact that at higher light intensity some species of microalgae show evidence of photodamage [31,38], so that Light intensity can become so high that it generates saturation or photoinhibition, causing microalgae growth inhibition [39]. However, it is important to mention that each microalgal morphotype has its own metabolic plasticity, so it is important to evaluate them and optimize variables such as light and photoperiod to know the metabolic efficiency in terms of lipid production, in this sense, the results indicate that the kinetic model can realistically re-flect the light energy utilization efficiency of microalgae as well as their intrinsic growth kinetic characteristics [22, 40].
- Moreover, there are editorial errors: In line 158 and 219 it should be 6 x 106 cells/ml instead of 6X106 cells; in various places in the text, the unit of light intensity is lux, lx and Lux; ml and mL. The text should be thoroughly revised and errors of this type should be corrected.
We agree with the reviewer and correct the errors raised above.
- Also, Figure 3 is not very clear - I suggest modifying it and adding standard error bars. The description is also about averages (± standard error) - no such data in the graph. In the graph, the colors represent the photoperiods, and the legend on the right adds no additional information.
We agree with the reviewer, therefore, we have eliminated graph 3 and table 2 has been improved, which shows the results of the yield of lipids acquired from the microalgae biomass.

Reviewer 3 Report
This authors needs to address some concerns:
How the authors selected this species of algae? I cant see the proper justification of selection in the introduction. The focus of the introduction should be more on this aspect not only general information.
It is recommended to add some experimental images to enhance the quality of the manuscript and make it more understandable.
How about biodiesel production? I suggest adding biodiesel results too as the paper now is too short and not too novel. The authors should evaluate the feasibility of biodiesel production in terms of an economic point of view?
How about lipid content? is it more than other species of algae or not?
The number of references for this paper is too many. You should reduce it to more updated references.
Author Response
Tuxtla Gutierrez Chiapas, Mexico, 25-04-2021
Dear Prof. Dr. Marc A. Rosen
Editor-in-Chief
Sustainability
We would like to submit the revised manuscript entitled ‘Influence of Light Intensity and
Photoperiod on the Photoautotrophic Growth and Lipid Content of the Microalgae
Verrucodesmus verrucosus in a Photobioreactor’ by Laura Velez et al. for publication in Sustainability. We corrected the manuscript and included all suggestions and comments made by the editor in the revised manuscript. Please find below the comments by the reviewers in italic, our answers in normal font and the changes to the text in red color.
We hope this makes the revised manuscript acceptable to be published in Sustainability.
Hoping to hear from you soon
Sincerely yours,
Dra. Yazmin Sánchez Roque
On behalf of the co-authors
Universidad Politécnica de Chiapas
e-mail: ysanchez@ia.upchiapas.edu.mx
_______________________________________________________
Reviewer 3
- How the authors selected this species of algae? I cant see the proper justification of selection in the introduction. The focus of the introduction should be more on this aspect not only general information.
We agree with the reviewer, therefore, we improved the "Algal sample" the methodology section
Line 97. “2.2. Algal Sample
The algae sample Verrucodesmus verrucosus was provided by the laboratory of applied science in Universidad Autónoma Metropolitana Iztapalapa (UAM). The species was re-cently described by Hegewald et al. [25]; López-Mendoza et al. [26] [27,28] and with Gen-Bank number JQ240289”.
- How about biodiesel production? I suggest adding biodiesel results too as the paper now is too short and not too novel. The authors should evaluate the feasibility of biodiesel production in terms of an economic point of view?
We appreciate the reviewer's comment, however, so far we have the results shown in the present work, but we consider that this research is of importance for future research for the production of biodiesel, in addition, the concentration of fatty acids is compared in function of light intensities, finally it is important to mention that V. verrucosus is a new microalgae with metabolic potential for future research.
We are grateful to review Table 2, Figure 2 and Table 3, to observe the new adjustments, with the aim of increasing the scientific contribution of this research work
- How about lipid content? ¿is it more than other species of algae or not?
We agree with the reviewer, so we include other findings related to microalgae corresponding to the same family of V. verrucosus (Scenedesmaceae).
Line 321- 344: Regarding the accumulation of lipids Keil et al. [41] and Mutaf et al. [42] they men-tions that green algae can accumulate lipids from 12.05 to 71.33% of fatty acids. The high concentration of lipids in green microalgae is related to the physicochemical composition of the substrate, so in this research work the BG11 medium was used which according to Sánchez et al. [24], this culture medium contains a concentration of nitrogen, less than 2mg / L, a concentration equivalent to that used by Xin et al. [43] who mention that under conditions of nitrogen limitation (2.5 mg L-1) or phosphorus (0.1 mg L-1), Scenedesmus sp. could accumulate lipids up to 53% [44-49].
In the present research work it is observed that the composition of fatty acids present was 64.29% correspond to saturated fatty acids (Table 2), as demonstrated by Tan et al. [50] they observed that the quality of biodiesel improved to greater saturation of fatty acids (typically in C16: 0 and C18: 0). The composition of saturated fatty acids in high concen-tration in green microalgae such as V. verrucosus is due to the low presence of the enzyme diacylglycerol acyltransferase responsible for the biosynthesis of neutral lipids [44].
So too, other researchers who have evaluated various microalgae corresponding to the Chlorophyceae class, specifically Scenedesmus sp. is part of the Scenedesmaceae fam-ily like V. verrucosus, this morphotype accumulated the 35-50% of lipids. On the other hand, lipid content is higher than Ettlia sp, Chlorella vulgaris, Neochloris oleabundans, Nan-nochloropsis sp, N. salina, N. oceanica, N. oculata and Scenedesmus dimorphus. Palmitic acid was identified as fatty acid more abundant with 40% this result indicated that it can be a promising feedstock for biodiesel production, because the most common components of biodiesel are palmitic acid, palmitoleic acid and oleic acid [24, 51-53].
- The number of references for this paper is too many. You should reduce it to more updated references.
We appreciate the reviewer's observation, however, the references were reviewed considering the most recent.

Round 2
Reviewer 1 Report
The authors have revised the MS adequately
Author Response
Tuxtla Gutierrez Chiapas, Mexico, 22-05-2021
Dear Reviewer
Sustainability
We would like to submit the revised manuscript entitled ‘Influence of Light Intensity and Photoperiod on the Photoautotrophic Growth and Lipid Content of the Microalgae Verrucodesmus verrucosus in a Photobioreactor’ by Laura Velez et al. for publication in Sustainability. We greatly appreciate your comments as you contribute to the improvement of the final manuscript.
Sincerely yours,
Dra. Yazmin Sánchez Roque
On behalf of the co-authors
Universidad Politécnica de Chiapas
e-mail: ysanchez@ia.upchiapas.edu.mx
Reviewer 2 Report
The authors revised the manuscript mostly in line with the comments. However, some minor adjustments are still needed.:
- if the composition of culture medium is given in the table, it does not need to be repeated in the text. It is enough to say that culture medium with the composition given in Table 1 was used,
- figure 2 is now easier to interpret, but if possible it would be preferable to give it a better quality,
- the conclusions require rewording - In the conclusions, the authors wrote that: „the Verrucodesmus verrucosusmicroalgae has a better growth and adaptation” – better than what? It would be also beneficial to complete them (at the beginning) with a brief description of what has been done, for which conditions / variables and the most important observations.
Author Response
Tuxtla Gutierrez Chiapas, Mexico, 17-05-2021
Dear Prof. Dr. Marc A. Rosen
Editor-in-Chief
Sustainability
We would like to submit the revised manuscript entitled ‘Influence of Light Intensity and
Photoperiod on the Photoautotrophic Growth and Lipid Content of the Microalgae
Verrucodesmus verrucosus in a Photobioreactor’ by Laura Velez et al. for publication in Sustainability. We corrected the manuscript and included all suggestions and comments made by the editor in the revised manuscript. Please find below the comments by the reviewers in italic, our answers in normal font and the changes to the text in red color.
We hope this makes the revised manuscript acceptable to be published in Sustainability.
Hoping to hear from you soon
Sincerely yours,
Dra. Yazmin Sánchez Roque
On behalf of the co-authors
Universidad Politécnica de Chiapas
e-mail: ysanchez@ia.upchiapas.edu.mx
_____________________________________________________
Manuscript 1192551 “Influence of Light Intensity and Photoperiod on the Photoautotrophic Growth and Lipid Content of the Microalgae Verrucodesmus verrucosus in a Photobioreactor”; Sustainability
REVIEWER 2
- If the composition of culture medium is given in the table, it does not need to be repeated in the text. It is enough to say that culture medium with the composition given in Table 1 was used,
We appreciate the editor's comments, so, we include the suggested wording and eliminate the composition of the medium present in the text.
Line: 103
The culture BG11 medium with the composition given in Table 1 was used, the pH was adjusted to 7.5, It was brought to final volume of 1 L at 100 °C [29] (Table 1).
- Figure 2 is now easier to interpret, but if possible it would be preferable to give it a better quality.
We agree with the reviewer's observation, therefore we have improved the quality of the image and the original version was included.
Line: 241
- The conclusions require rewording - In the conclusions, the authors wrote that: „the Verrucodesmus verrucosus microalgae has a better growth and adaptation” – better than what?
We appreciate the editor's comments, so, we restructure the conclusion based on your comment, to show that the comparison is related to green microalgae.
Line: 345 to 358
In the present research work, the metabolic plasticity of microalgae is corroborated, demonstrating that the conditions to which they are exposed determine the accumulation of lipids, confirming the potential of green microalgae corresponding to the Chlorophyceae class as the group that has reported greater concentration of lipids [53], in this sense the results showed that the Verrucodesmus verrucosus microalgae has a better growth and adaptation in a 3L tubular photoreactor under controlled conditions at a light inten-sity of 2000 Lux and a 12L: 12D photoperiod, with a total lipid accumulation of 50. 42% with fatty acids such as palmitic acid, oleic acid, stearic acid, palmitoleic acid and myristic ac-id in concentrations of 44.90, 25.81, 9.81, 7.87 and 5.93% respectively.
However, it is important to mention that each microalgae is specific, so it requires a previous evaluation that allows the optimization of the conditions to potentiate the pro-duction of lipids that can be used for conversion to biodiesel. This specificity turns out to be very promising for future research with morphotypes that, in addition to achieving high yields, it can be produced at low cost.
- It would be also beneficial to complete them (at the beginning) with a brief description of what has been done, for which conditions / variables and the most important observations.
We agree with the reviewer, therefore, for a better understanding, the introduction has been restructured, including results from other authors who evaluated variables on green microalgal biomass for lipid production.
Line: 62 to 70
Microalgae have metabolic plasticity that allows them to adapt to different eco-sys-tems and biotechnological processes, generating biomass that can be used in the produc-tion of food, concentrates, bioactive compounds, in bioremediation and biofuels. Since light, aeration, pH, light type, photoperiod and temperature (1500 Lux, 5% of CO2, 7.9, blue at 475nm, 16L:08D and 22°C respectively) in average are the main determinants of biomass productivity of Chlorophyceae microalgae, the largest accumulators of lipids so far, of crops in open-air ponds operated under well-mixed basic nutrients (N, and P), it is necessary to know the specific growth rate of the strain selected as a function of these two variables [1,2,5,7,8,15,16].

Reviewer 3 Report
The paper is not strong enough as an academic paper. It has many mistakes and shortcomings and a lack of analysis and justifications. Unfortunately, it is not ready for publication due the following reasons:
- the novelty is not clear. Why the author choose this species of algae?
- The language is too poor in whole the manuscript.
- line 328: how the authors cited 5 references in one sentence? All of them found the same results?
- Except Table 1 and Figure 1 which show general information, the manuscript only has 1 Table and 2 Figures for the results. I think is not enough for an academic paper.
- The conclusion is too short and not impressive.
- References are too many.
- Biodiesel results need to be added to make the manuscript stronger. Now, the data are not enough.
Author Response
Tuxtla Gutierrez Chiapas, Mexico, 17-05-2021
Dear Prof. Dr. Marc A. Rosen
Editor-in-Chief
Sustainability
We would like to submit the revised manuscript entitled ‘Influence of Light Intensity and
Photoperiod on the Photoautotrophic Growth and Lipid Content of the Microalgae
Verrucodesmus verrucosus in a Photobioreactor’ by Laura Velez et al. for publication in Sustainability. We corrected the manuscript and included all suggestions and comments made by the editor in the revised manuscript. Please find below the comments by the reviewers in italic, our answers in normal font and the changes to the text in red color.
We hope this makes the revised manuscript acceptable to be published in Sustainability.
Hoping to hear from you soon
Sincerely yours,
Dra. Yazmin Sánchez Roque
On behalf of the co-authors
Universidad Politécnica de Chiapas
e-mail: ysanchez@ia.upchiapas.edu.mx
________________________________________________
Manuscript 1192551 “Influence of Light Intensity and Photoperiod on the Photoautotrophic Growth and Lipid Content of the Microalgae Verrucodesmus verrucosus in a Photobioreactor”; Sustainability
REVIEW 3
- The novelty is not clear. Why the author choose this species of algae?
We appreciate the reviewer's observation, so below we explain the reasons why the morphotype Verrucodesmus verrucosus was used.
In the commercial production of microalgae, the adaptation of the strain to different living conditions is key, in such a way that it is able to resist and redirect the metabolism towards the accumulation of lipids that can be used for the production of biodiesel, in this sense There is ample potential for those morphotypes that have not yet been sufficiently evaluated, such as V. verrucosus, this microalga corresponds to genus chlorophycea that is characterized by rapid growth in cell culture, so it has the ability to use organic and inorganic compounds as a nutritional substrate [28-31].
“Verrucodesmus verrucosus” is a species that has not been sufficiently reported in the bioenergetics area, so far the metabolic efficiency for the accumulation of lipids at the flask level has been reported (Sánchez-Roque 2020), however it is important to report as impact the light on this morphotype and how it behaves in a larger-scale production system, results that are observed in the present research.
- line 328: how the authors cited 5 references in one sentence? All of them found the same results?
We appreciate the reviewer's observation, however the references used support the wording based on the importance of the chemical composition of the culture medium and the morphotype of microalgae to define its ability to accumulate lipids that can be extracted and transesterified to produce biodiesel.
- Except Table 1 and Figure 1 which show general information, the manuscript only has 1 Table and 2 Figures for the results. I think is not enough for an academic paper.
We appreciate the reviewer's observations, so we have implemented another table and image to strengthen the results in the latest version of the document.
Figure 2 shows the behavior of the microalgal biomass through the behavior of the variables that define the replication speed of the biomass, thus also demonstrating in the Table 2 the accumulation of lipids in relation to nitrogen consumption, under the influence of different photoperiods and Light intensity.
Figure 3 shows the profile of fatty acids of the microalgal biomass under the influence of different light intensities, corresponding to the treatments that allowed the greatest accumulation of lipids, finally table 3 shows the abundance and type of fatty acids identified in the chromatogram.
These findings undoubtedly provide relevant information for future research, aimed at improving the production of quality lipids for the production of biodiesel.
- The conclusion is too short and not impressive.
We agree with the reviewer's observation, in this sense the conclusion was restructured to allow a better understanding of the achievements of the present research work.
Line: 345 to 358
In the present research work, the metabolic plasticity of microalgae is corroborated, demonstrating that the conditions to which they are exposed determine the accumulation of lipids, confirming the potential of green microalgae corresponding to the Chlorophyceae class as the group that has reported greater concentration of lipids [53], in this sense the results showed that the Verrucodesmus verrucosus microalgae has a better growth and adaptation in a 3L tubular photoreactor under controlled conditions at a light intensity of 2000 Lux and a 12L: 12D photoperiod, with a total lipid accumulation of 50. 42% with fatty acids such as palmitic acid, oleic acid, stearic acid, palmitoleic acid and myristic ac-id in concentrations of 44.90, 25.81, 9.81, 7.87 and 5.93% respectively.
However, it is important to mention that each microalgae is specific, so it requires a previous evaluation that allows the optimization of the conditions to potentiate the pro-duction of lipids that can be used for conversion to biodiesel. This specificity turns out to be very promising for future research with morphotypes that, in addition to achieving high yields, it can be produced at low cost.
- References are too many.
We appreciate the comments of the reviewer, however we have reviewed the references and note that all were used for the development and analysis of this research work.
- Biodiesel results need to be added to make the manuscript stronger. Now, the data are not enough.
We appreciate the reviewer's observation, however we do not have results of the transesterification process, but according to other investigations, those who have evaluated a green microalgae the obtained yields of the transesterification reactions were the 87.4% to 94.6%. The methyl esters contained higher percentage of saturated fatty acids and its fatty acid composition improved the basic fuel properties. The overall finding of the study suggested that class Chlorophyceae can be considered as more viable source for biodiesel production, in part due to the relative ease in producing a high oil yield [41,42,44,51-53].
- How about biodiesel production? I suggest adding biodiesel results too as the paper now is too short and not too novel. The authors should evaluate the feasibility of biodiesel production in terms of an economic point of view?
We appreciate the reviewer's observation, however we do not include the feasibility analysis of biodiesel production from the economic point of view because it does not correspond to the objective of the article, however, in response to your request, we carried out the analysis according to our experience in this line of research for the generation of biodiesel, such information is detailed below.
As preliminary studies for this work, biodiesel has been produced from various sources, which include vegetable oil from Jatropha curcas and Ricinus communis, as well as from used cooking oils (WCO) (Berrones-Hernández et al. 2019; Berrones-Hernández et al. 2020 Sánchez-Roque et al. 2020) With the interest of studying the economic factor of biodiesel production from various raw materials, a pilot plant was installed in the Agroindustrial Engineering workshop of the Universidad Politécnica de Chiapas, in which biodiesel production costs have been estimated for each liter of oil fed to the reactor. The raw material used in the cost estimate was WCO. The pilot plant has a production capacity of 60 liters of biodiesel per batch. The following table shows the actual data for production costs.
Supplies |
Quantity |
Unit |
Unit cost |
Cost per liter |
WCO LAB plant oil |
1 |
l |
$ 1.20 |
$ 1.20 |
Freight and fuel cost |
1 |
Service |
$ 3.20 |
$ 3.20 |
Electric energy (Includes electric motors for agitators, pumps, and minor energy such as lighting and computer equipment) |
1 |
Service |
$ 0.17 |
$ 0.17 |
Gas L.P. for oil heating |
1 |
Service |
$ 0.16 |
$ 0.16 |
Methanol |
0.2125 |
l |
$ 12.61 |
$ 2.68 |
Operation and maintenance |
2 |
Operators |
$ 1.87 |
$ 3.74 |
Catalyst (NaOH) |
5.3 |
g |
$ 0.07 |
$ 0.37 |
Amortization |
1 |
Service |
$ 0.48 |
$ 0.48 |
Production cost |
$ 12.00 |
|||
Profit (34%) / liter |
$ 4.08 |
|||
Sales cost without taxes |
$ 16.08 |
Now the production of microalgae is being studied as a source of raw material for feeding the pilot plant, in order to estimate the real costs of production. From the data derived from this research, it is known that for each kilogram of microalgae biomass a yield of 50.42% of total oil is obtained, which exceeds those reported by other authors in the production of continuous tubular reactors (Stefano, 2020). However, in the experimental stage, the feasibility is still low. Some authors report production costs of € 4.15 ($ 100.38 Mexican pesos) per kilogram of microalgae, in dry weight (DW) (Norsker et al. 2011). If we consider these production costs and the oil yield per kilogram of dry matter and introduce them to the cost analysis in the table above, the pilot scale production cost would be as shown below:
Supplies |
Quantity |
Unit |
Unit cost |
Cost per liter |
LAB plant microalgae oil |
1 |
l |
$ 199.09 |
$ 199.09 |
Electric energy (Includes electric motors for agitators, pumps, and minor energy such as lighting and computer equipment) |
1 |
Service |
$ 0.17 |
$ 0.17 |
Gas L.P. for oil heating |
1 |
Service |
$ 0.16 |
$ 0.16 |
Methanol |
0.2125 |
l |
$ 12.61 |
$ 2.68 |
Operation and maintenance |
2 |
Operators |
$ 1.87 |
$ 3.74 |
Catalyst (NaOH) |
5.3 |
g |
$ 0.07 |
$ 0.37 |
Amortization |
1 |
Service |
$ 0.48 |
$ 0.48 |
Production cost |
$ 206.69 |
|||
Profit / liter (34%) |
$ 70.28 |
|||
Cost of sale |
$ 276.97 |
With this analysis and considering the same production criteria, it can be observed that the cost of sale of biodiesel from microalgae rises 16 times more compared to the production from WCO. This difference is related to different factors such as the disposition of raw materials, one is an easily recoverable waste, while the microalgal material must be produced, this fact represents an interesting area of ​​opportunity since it allows to potentiate the metabolic plasticity of these microorganisms to improve conventional biodiesel production processes.
Another important factor is the scale of production. A laboratory tubular reactor is a test reactor, which in this study is intended to find operating conditions that favor the development of microalgae morphotypes and oil production yields, for which the present work provides results interesting aimed at identifying the best production conditions that benefit production performance, to later evaluate larger-scale reactors to increase production performance and optimize costs.
- Sánchez-Roque, Y., Luna, Y. P., Acosta, J. M., Vázquez, N. F., Sebastian, J. P., & Hernández, R. B. (2020). Optimization for the production of verrucodesmus verrucosus biomass through crops in autotrophic and mixotrophic conditions with potential for the production of biodiesel. Revista Mexicana de Ingeniería Química, 19(1), 133-147.
- Berrones-Hernández, R., del Carmen Pérez-Luna, Y., Sánchez-Roque, Y., Pantoja-Enríquez, J., Grajales-Penagos, A. L., López-Cruz, C. F., ... & Sebastian, P. J. (2019). Heterogeneous esterification of waste cooking oil with sulfated titanium dioxide (STi). BioEnergy Research, 12(3), 653-664.
- Berrones Hernández, R. (2019). Catálisis heterogénea con catalizadores óxido-metálicos ZrO2 y TiO2 en los procesos de producción de Biodiésel (Master's thesis, Universidad de Ciencias y Artes de Chiapas-Instituto de Investigación e Innovación en Energías Renovables-Doctorado en Materiales y Sistemas Energéticos Renovables).
- Berrones-Hernández, R., Trejo-Hernández, G., del Carmen Pérez-Luna, Y., Sánchez-Roque, Y., Rojas-Blanco, L., Zamudio-Torres, I., ... & Ramírez-Morales, E. (2020). Catalytic activity of Srilankite nanoparticles in the esterification of oleic acid.
- Norsker, N. H., Barbosa, M. J., Vermuë, M. H., & Wijffels, R. H. (2011). Microalgal production—a close look at the economics. Biotechnology advances, 29(1), 24-27.
- Stefano, F. (2020). Innovative Plant for the Production of Microalgaes. Asian Journal of Applied Science and Technology, 4(3), 145-166.

Round 3
Reviewer 3 Report
The paper is improved but still needs consideration:
- The language is poor. Some parts are written in the past form and some of them in the present form.
- the weak section is the introduction. from the beginning, the authors discussed related to biodiesel production while they don't have any results in this content. Even the cited references are old and show that the authors didn't read the recent studies for biodiesel of production from microalgae. studies like: Muhammad, G., Alam, M. A., Mofijur, M., Jahirul, M. I., Lv, Y., Xiong, W., ... & Xu, J. (2021). Modern developmental aspects in the field of economical harvesting and biodiesel production from microalgae biomass. Renewable and Sustainable Energy Reviews, 135, 110209. Ananthi, V., Brindhadevi, K., Pugazhendhi, A., & Arun, A. (2021). Impact of abiotic factors on biodiesel production by microalgae. Fuel, 284, 118962. Rezania, S., Oryani, B., Park, J., Hashemi, B., Yadav, K. K., Kwon, E. E., ... & Cho, J. (2019). Review on transesterification of non-edible sources for biodiesel production with a focus on economic aspects, fuel properties and by-product applications. Energy Conversion and Management, 201, 112155.
- lines 71-74 are not acceptable as the reference is not valid.
- where is the explanation of Figure 3?
- All the added references which are in red are not acceptable. You should refer to English, recent, and high-quality papers only. Even the majority of references are old. The authors should consider now is the year 2021.
- Usually, we should not have citations in the conclusion.
Author Response
Tuxtla Gutierrez Chiapas, Mexico, 22-05-2021
Dear Prof. Dr. Marc A. Rosen
Editor-in-Chief
Sustainability
We would like to submit the revised manuscript entitled ‘Influence of Light Intensity and
Photoperiod on the Photoautotrophic Growth and Lipid Content of the Microalgae
Verrucodesmus verrucosus in a Photobioreactor’ by Laura Velez et al. for publication in Sustainability. We corrected the manuscript and included all suggestions and comments made by the editor in the revised manuscript. Please find below the comments by the reviewers in italic, our answers in normal font and the changes to the text in red color.
We hope this makes the revised manuscript acceptable to be published in Sustainability.
Hoping to hear from you soon
Sincerely yours,
Dra. Yazmin Sánchez Roque
On behalf of the co-authors
Universidad Politécnica de Chiapas
e-mail: ysanchez@ia.upchiapas.edu.mx
_____________________________________________________________
Manuscript 1192551 “Influence of Light Intensity and Photoperiod on the Photoautotrophic Growth and Lipid Content of the Microalgae Verrucodesmus verrucosus in a Photobioreactor”; Sustainability
REVIEWER 3
- The language is poor. Some parts are written in the past form and some of them in the present form.
We appreciate the editor's comments, so, we review and improve both the grammar and the writing of the manuscript.
- The weak section is the introduction. from the beginning, the authors discussed related to biodiesel production while they don't have any results in this content. Even the cited references are old and show that the authors didn't read the recent studies for biodiesel of production from microalgae.
We agree with the reviewer's observations, so we have restructured and strengthened the introduction with the references provided and other research papers.
Line: 30-78
- Introduction
Biofuel production and its research is an issue of global importance due to ever-increasing concerns over the depletion of fossil fuels and their devastating impact on the environment, at present, more attention has been given to the production of biodiesel [1, 2] which is a liquid biofuel, obtained from a transesterification process of an oil or fat with an alcohol [3]. The most common raw materials that are used for the production of biodiesel have been the basic vegetable oils, however, this represents multiple problems such as the use of basic oils for nutritional purposes, the use of agricultural land, the use of pesticides and fertilizers necessary for crops, which in turn filter out contaminating the groundwater [4, 2]. One of the potentially renewable alternative raw materials for the production of lipids to produce biodiesel is microalgae, with the potential to overcome many limitations, they are capable of satisfying the world demand for transportation fuels [5] and they seem to be the only source to completely displace diesel fossil [6]. Microalgae are photosynthetic microorganisms [7] with simple growth requirements that produce lipid around 80% by weight of dry mass, but the levels commonly obtained are 20-50% [8].
Unlike terrestrial crops, microalgae grow 10-50 times faster [5], which leads to higher biofuel volumes per hectare, for example the yield of microalgae oil (58,700 L ha-1) is about 31 times higher than that of Jatropha (1, 892 L ha-1) [2]. On the other hand, for the cultivation of microalgae, there are different systems [9], which use light (as an energy source), water and carbon dioxide (as a carbon source) to cultivate microorganisms, these in turn perform photosynthesis to generate biomass and oxygen [10]. Therefore, illumination factors, such as light intensity [11], photoperiod duration [12] and wavelength play an important role in the process of photosynthesis and affect photoautotrophic growth [13] and lipid content [14] of microalgae.
Specifically, photobioreactors currently use artificial light to characterize and optimize the living conditions of a microalgal morphotype, the most widely used are the Light emitting diodes (LEDs) have become the currently preferred artificial light source in most phototrophic cultivations because of their durability (lifetime> 25-50,000h) and lack of toxic elements such as mercury, unlike fluorescent lamps [15]. LEDs have also the advantage of being fast-responding diodes emitting nearly mono- or multichromatic light at desired wavelengths, thus being ideal for studies on the light requirements of microalgae for growth and light-dependent induction of specific target metabolites, without forgetting the importance of the times of exposure to light called photoperiods [16]. Phototrophic cultivations benefit from high power conversion efficiencies of LEDs, transforming more than 40–50% of electrical energy into light that can be effectively utilized by phototrophs for photosynthesis. However, it is important to consider the optimal specificity of each microalgae to obtain the metabolites of interest, so its evaluation must be particular [17].
Due to the aforementioned, the Microalgaes have metabolic plasticity that allows them to adapt to different ecosysitems and biotechnological processes that must be characterized, so far it is known that the main average variables that determine the accumulation of lipids in the class Chlorophyceae They are the light, aeration, pH, light type, photoperiod and temperature, with reported records of 1500 Lux, 5% of CO2, 7.9, blue at 475nm, 16L: 08D and 22 ° C, respectively, the evaluations on this class of microalgae has reported the highest levels of accumulated lipids. [1,9,11,13,16,17].
For this reason, in this work focuses on investigating the effect of light intensity and photoperiod on the growth of the microalga Verrucodesmus verrucosus using as a light source blue LED (465-470 nm), in a batch culture in a column photobioreactor design with volume of 3 L.
Also, to comment that according to their appreciable observations, we reviewed current research works, which allowed us to strengthen the manuscript and update the references.
Line: 376-471
- Lines 71-74 are not acceptable as the reference is not valid.
We agree with the reviewer's observation, therefore we have eliminated the information contained in lines 71 to 74
- Where is the explanation of Figure 3?
We appreciate the reviewer's observation, so we have improved the wording of the methodology and the description of Figure 3 has been included.
Line: 218-227
With regard to lipid production, it was observed that during on flask evaluation, the maxi-mum production was 24.31%, however a significant difference was observed under the evaluation in a photobioreactor at different light inten-sities and photoperiods, observing the best results at 2000 Lux and photoperiod 12:12 where the lipid concentration doubled (50.42%), this high concentration of lipids is related to the high consumption of nitrogen (66%) by microalgae biomass(Table 2), On the other hand, the lower production of lipid (13. 61%) was identified at 1000 Lux with photoper-iod 16:08.
Regarding the chromatographic analysis of the profile of fatty acids present in the lipid samples obtained from the better yields at different light intensities, palmitic acid was observed as the most abundant of 42.03 to 55.21% (Figure 3). Of the identified fatty acids, 64.29% correspond to saturated fatty acids and 35.71% correspond to monounsaturated fatty acids (Table 3).
- All the added references which are in red are not acceptable. You should refer to English, recent, and high-quality papers only. Even the majority of references are old. The authors should consider now is the year 2021.
We appreciate the reviewer's observation, so we have reviewed research that allowed us to update and improve the quality of the information concentrated in the manuscript.
Line: 374-470
- Usually, we should not have citations in the conclusion.
We appreciate the reviewer's observation, therefore, we remove the reference from the conclusión.
Line: 343-357

Round 4
Reviewer 3 Report
The paper is acceptable after checking the language carefully.